# Antibacterial Activity of Co(III) Complexes with Diamine Chelate Ligands against a Broad Spectrum of Bacteria with a DNA Interaction Mechanism

**DOI:** 10.3390/pharmaceutics13070946

**Published:** 2021-06-24

**Authors:** Katarzyna Turecka, Agnieszka Chylewska, Michał Rychłowski, Joanna Zakrzewska, Krzysztof Waleron

**Affiliations:** 1Department of Pharmaceutical Microbiology, Faculty of Pharmacy, Medical University of Gdańsk, gen. Hallera 107, 80-416 Gdańsk, Poland; krzysztof.waleron@gumed.edu.pl; 2Department of Bioinorganic Chemistry, Faculty of Chemistry, University of Gdańsk, Wita Stwosza 63, 80-308 Gdańsk, Poland; agnieszka.chylewska@ug.edu.pl; 3Laboratory of Virus Molecular Biology, Intercollegiate Faculty of Biotechnology, University of Gdańsk and Medical University of Gdańsk, Abrahama 58, 80-307 Gdańsk, Poland; michal.rychlowski@biotech.ug.edu.pl; 4Centre of Molecular and Macromolecular Studies, Polish Academy of Sciences, Henryka Sienkiewicza 112, 90-001 Łódź, Poland; jzakrzew@cbmm.lodz.pl

**Keywords:** Co(III) coordination complexes, antibacterial activity, microbroth dilution method, minimum inhibitory concentration, minimum bactericidal concentration, synergy assay, serial passages assay, DNA interactions

## Abstract

Cobalt coordination complexes are very attractive compounds for their therapeutic uses as antiviral, antibacterial, antifungal, antiparasitic, or antitumor agents. Two Co(III) complexes with diamine chelate ligands ([CoCl_2_(dap)_2_]Cl (**1**) and [CoCl_2_(en)_2_]Cl (**2**)) (where dap = 1,3-diaminopropane, en = ethylenediamine) were synthesized and characterized by elemental analysis, an ATR technique, and a scan method and sequentially tested against Gram-positive and Gram-negative bacteria. The minimum inhibitory concentration results revealed that anaerobic and microaerophilic bacteria were found to be the most sensitive; the serial passages assay presented insignificant increases in bacterial resistance to both compounds after 20 passages. The synergy assay showed a significant reduction in the MIC values of nalidixic acid when combined with Compounds (**1**) or (**2**). The assessment of cell damage by the complexes was performed using scanning electron microscopy, transmission electron microscopy, and confocal microscopy, which indicated cell membrane permeability, deformation, and altered cell morphology. DNA interaction studies of the Co(III) complexes with plasmid pBR322 using spectrophotometric titration methods revealed that the interaction between Complex (**1**) or (**2**) and DNA suggested an electrostatic and intercalative mode of binding, respectively. Furthermore, the DNA cleavage ability of compounds by agarose gel electrophoresis showed nuclease activity for both complexes. The results suggest that the effect of the tested compounds against bacteria can be complex.

## 1. Introduction

Increased antimicrobial resistance to the current antibiotics among bacteria is a serious problem globally. To overcome this problem, new and effective antibacterial compounds with low toxicity are needed. A promising group of antimicrobial compounds seem to be metallopharmaceuticals due to their abundance and structural diversity. The most remarkable achievements in the research on this type of compound refer to d-block metal-based drugs. D-block metal ion complexes encounter a large number of biomolecules in biological systems (amino acids, proteins, oligonucleotides, or DNA) and can interact with them. Comprehensive in vitro studies (evaluations of the structure of the complex, including detailed knowledge of bond length, the angles between them, their energy, the acid–base nature, and behaviour in the aquatic and non-aqueous environment) are necessary to design a metal drug or metal pro-drug useful for treatment. The kinetic and thermodynamic stability of metal-based pharmaceuticals against pH changes, temperature, chemical agents, the resistance of compounds to metabolism, the hydrophilic–lipophilic nature or ionization, and particle size is very important in drug development as well [1]. The growing interest of researchers in this type of compound is due to ligand complexes of transition metals resulting in antibacterial, antifungal, antiviral, antitumor, and anti-inflammatory properties [2,3,4,5,6,7,8]. There have been many reports on the use of metal-containing drugs. Complexes of platinum (cisplatin), gold (auranofin used in therapy of rheumatoid arthritis), rhenium (radiopharmaceutical used in imaging and radiotherapy), ruthenium (anticancer drugs) or cobalt, lithium, bismuth, iron, calcium, copper, and zinc have been used in medicine [9,10,11,12,13,14]. Recently, the interest in cobalt coordination complexes, especially Co(III) complexes, due to their antimicrobial and antitumor properties has grown [15,16,17,18,19,20,21,22,23,24].

The main feature allowing the use of cobalt(III) complexes as components of chemotherapeutic agents is the presence of stable Co(III) and labile Co(II). Complexes of Co(II) are stable in solid form but present a remarkable ease of oxidation under biological conditions. The antibacterial properties of Co(III) complexes have been frequently reported, and the increased effectiveness of cobalt ion coordination to the appropriate ligand when compared to the free ligand itself has been emphasized. It was confirmed that the values of the electrochemical redox potential of the complex and the structure of the ligand have a significant impact on the effectiveness and the stability of compounds used [25,26]. Optical isomers of Co(III) with ethylenediamine as *N*,*N*-bidentate ligand [Co(en)_3_](NO_3_)_3_ have been studied and have exhibited high antimicrobial activity, irrespective of the isomer used [27]. Coordination compounds of cobalt(III) with pyridine-amide tri- or bidentate ligands have also been studied [28] and presented strong antibacterial properties for *Escherichia coli*, *Shigella flexneri*, and *Klebsiella planticola*. These complexes were also tested for the cytotoxic activity on the HEK cell lines and showed less cytotoxicity compared with the commercial antibiotic (gentamycin) at all tested concentrations.

There is a large number of cobalt(III) complex compounds that also exhibit potent antiviral activity, such as the CTC-96 (CTC series is a class of Co(III) complexes containing N, O-donor ligands), a potent agent against herpes simplex virus type 1 (HSV-1) [29]. This compound is present in the medication to prevent conjunctivitis, and it prevents epithelial corneal inflammation.

In the scientific literature, the anticancer activity of Co(III) complexes has also been described. Although the mechanism of this action has not been resolved yet, it is known that the important role in the antitumor activity is played by ligand activity, redox potential, and the different durability of Co^2+^ (unstable) ions and Co^3+^ (neutral and stable) ions. The active ligand is deactivated when combined with the cobalt atom. The cytotoxic agent is released due to the reduction of Co(III) to Co(II) under hypoxic conditions. Compounds investigated for this effect are cobalt (III) complexes with Schiff’s four-coarse bases, nitrogen mustards, other DNA alkylating agents, and MMP (extracellular matrix metalloproteinase) inhibitors. Measurements involving substances used in anticancer therapies consist of finding cytostatic factors that strongly interact with cancer cells and very weakly influence the body’s health cells [30,31].

The Co(III)-based antimicrobial and anticancer compounds can target DNA [32], block metal binding sites in enzymes [33], and interact with cytoplasmic proteins or other biomolecules [34].

In the search for new antibacterial agents, due to the growing resistance to antibiotics, the metallopharmaceuticals is a promising alternative. In the presented study, we determined the antibacterial activity of Co(III) complexes with simple bidentate inorganic ligands (en = ethylenediamine; dap = 1,3-diaminopropane) (Figure 1) on a broad spectrum of bacteria cultured in both aerobic and anaerobic conditions. The influence of the tested compounds on the bacterial DNA was also evaluated.

## 2. Materials and Methods

### 2.1. Materials and Growth Conditions

The reference and clinical strains of bacteria used in the study are listed in Table 1. The reference strains originated from the LGC standards. The clinical strains of bacteria from the Department of Pharmaceutical Microbiology collection were used. Strains were stored as glycerol stocks at −70 °C. For research purposes, cultures were conducted at 37 °C for 24 h in Mueller-Hinton (MH) broth, with 5% sheep blood (*S. pneumoniae* and *S. pyogenes*) and thioglycolate broth (*P. acnes* and *Clostridium* spp.) (Merck, Warsaw, Poland).

### 2.2. Synthesis of [CoCl_2_(N,N)_2_]Cl

All starting materials were commercially available (Sigma-Aldrich): cobalt(III) chloride hexahydrate, hydrochloride acid, hydrogen peroxide (30%), ethylenediammine, 1,3-diamminepropane, and methanol. All the solutions under study were prepared with twice-distilled water (Hydrolab-Reference purified) with conductivity not exceeding 0.09 µS/cm. The synthesis and physicochemical characterization of [CoCl_2_(dap)_2_]Cl (**1**) and [CoCl_2_(en)_2_]Cl (**2**) were performed according to the procedure described previously [35].

### 2.3. Synthesis of [Co(dap)_2_FLU]Cl_2_

A mixture of [CoCl_2_(dap)_2_]Cl·2H_2_O (1.05 g, 3 mmol) and fluorescein sodium salt (FLU) (1.51 g; 4 mmol) was dissolved in ethanol (6 mL), and NaBr (0.30 g; 3 mmol) in H_2_O was added (5 mL). The mixture was heated in a water bath until the final volume was reduced to half of its initial volume. It was then cooled on ice, and a dark violet solid was formed. The complex [Co (dap)_2_FLU]Cl_2_ was collected and recrystallized from H_2_O (20 mL). The yield through this method was ca. 60%. UV–vis: maxima at 231, 444, and 479 nm; ATR (cm^−1^): (NH amine) 3418 and 3191; (C=O) 1583; (C=C) 1470; (Co-N(dap)) 610; (Co-O (FLU)) 734; see Appendix A. Anal. calc. for C, 54.32; H, 5.61; N, 9.74; found C, 54.35; H, 5.57; N, 9.79.

### 2.4. Physicochemical Measurements

The structural analyses of Co(III) synthesized complexes were prepared by using three independent analytical techniques. To determine percentage compositions of the elements (C, H, N, S) of the synthesized compounds an element analyzer Carlo Erba EA 1108 CHNS (Elementar, Langenselbold, Germany) was used. The oscillatory spectra were recorded in the wavenumber range 4000–400 cm^−1^ on a Spectrum Two FT-IR instrument (Perkin Elmer, Waltham, MA, USA) using the ATR technique. ^1^H NMR spectra were recorded with a Brüker AVANCE 700 MHz spectrometer (Brüker, Billerica, MA, USA) at the NMR Laboratory at the Faculty of Chemistry (University of Gdańsk). Chemical shifts of proton NMR spectra were obtained by using two samples, FLU and [Co(dap)_2_FLU]Cl_2_, in the d^6^-DMSO as solvent. The XRD powder diffractograms of three Co(III) complexes were recorded by using a diffractometer to prove the synthetic products differences in their structures. The D2 Physier X-ray powder diffractometer (D2 Phaser model by Brüker) was used to collect the XRD patterns for the solid samples using CuKα1 radiation (measurement parameters: 2 Theta between 5–60. The scan method used a double beam mode with a zero/baseline correction. A jacketed titration cell connected to a constant temperature water bath set to 25.0 ± 0.1 °C was used to all mentioned spectroscopic measurements.

### 2.5. Determination of the Minimum Inhibitory Concentration (MIC) and Minimum Bactericidal Concentration (MBC)

The MIC and MBC of tested compounds were performed according to the procedure described by Turecka et al. [5], except that MH broth was used. Anaerobic bacteria were cultured using Genbaganaer (bioMerieux, Marcy l’Etoile, France).

### 2.6. Serial Passages Assay

Four strains of bacteria (*S. aureus* ATCC 6538, *S. aureus* MRSA/h-VISA 6347, *P. aeruginosa* ATCC 9027, and *P. aeruginosa* 12274) were selected for the passages studies. Assay was performed according to the procedure described by Turecka et al. [5], except that MH broth was used.

### 2.7. Synergy Assay

To determine the fractional inhibitory concentrations (FICs) of antibiotics (ampicillin, gentamicin, polymyxin B, and nalidixic acid) (Merck, Warsaw, Poland) in combination with Co(III) complexes, a checkerboard assay (CLSI) was used in 96-well microtiter plates. The concentration ranges for antibiotics were from 4096 to 0.125 µg/mL, and those for the Co(III) complexes were from 8000 to 125 µg/mL. All steps of the assay were performed according to the procedure described by Turecka et al. [5], except that MH broth was used. The results were interpreted as described in Meletiadis et al. (2005) and Odds (2003) [36,37].

### 2.8. Microscopic Analysis

#### 2.8.1. Transmission Electron Microscopy (TEM)

The suspension of *S. aureus* ATCC 6538 and *P. aeruginosa* ATCC 9027, cultured overnight in the MH broth, were diluted to 0.5 McFarland (with the same broth). The tested compounds in a concentration equivalent to MIC were introduced to 5 mL of the bacteria suspension and then incubated at 37 °C for 24 h. Following the incubation period and centrifugation, probes were fixed for 18 h in 2.5% glutaraldehyde (Polysciences, Warrington, PA, United States) in PBS (phosphate-buffered saline, pH 7.4), washed three times in the same buffer, and postfixed overnight in 1% osmium tetroxide (Polyscience, Warrington, PA, United States). Next, ethanol was dehydrated, and probes were embedded in Epon resin (Merck, Warsaw, Poland) and cut on the ultramicrotome Leica UC7 (Leica Microsystems, Wetzlar, Germany). Finally, probes were contrasted in uranyl acetate and lead citrate. A Philips CM100 transmission electron microscope (Philips, Amsterdam, the Netherlaands) was used [38].

#### 2.8.2. Scanning Electron Microscopy (SEM)

The potential effect of Co(III) complexes on the bacterial cell morphology using SEM was determined according to the procedure described by Khan et al. [39] for reference strains of *S. aureus* and *P. aeruginosa* cells. Scanning Electron Microscope JSM-6010LA by JEOL (Akishima, Tokyo, Japan), equipped with energy-dispersive X-ray spectrometer EDX, was used.

#### 2.8.3. Confocal Microscopy Assay

The cultures of *S. aureus* and *P. aeruginosa* were grown in LB broth until mid-log phase and then centrifuged. The bacterial pellets were washed three times using PBS buffer, pH 7.4, and resuspended in the same buffer to achieve the optical density corresponding to 10^6^ CFU/mL. Bacterial cells were incubated with FITC-labelled [Co(dap)_2_]Cl_2_ at MIC concentration and with fluorescein alone for 1 h at 37 °C. After incubation, the cells were centrifuged, washed three times with PBS buffer, and fixed on a glass slide. After that, the cells were evaluated using Confocal microscopes from Leica Microsystems (Wetzlar, Germany) and the images were produced. The same studies were performed for fluorescein alone.

### 2.9. Interactions of Compounds ***(1)*** and ***(2)*** with Bacterial DNA

#### 2.9.1. DNA Binding Studies of Diamine Co(III) Complexes

The electronic spectra were recorded on an Evolution 300 spectrophotometer (Thermo Fischer Scientific, Waltham, MA, USA) in the range of 200–700 nm in the case of UV detection of intermolecular interactions with biomolecules together with a spectral band with a 2 nm width. The binding ability of Co(III) complexes with diamines to DNA (plasmid pBR322) was investigated by spectrophotometric titration. The tris-(hydroxymethyl)-amino methane (Tris-HCl) buffer, pH 7.39, was used to prepare a solution of DNA. The concentration of freshly prepared DNA was calculated using an absorbance value at 258 nm according to the buffer solvent used (Appendix A) and the calibration curve, which indicates the DNA molar absorption coefficient (ε_DNA_ 6600 M^−1^ cm^−1^) [40,41]. The spectrophotometric titrations were performed at room temperature by gradually increasing the DNA concentration and constant concentrations of the complexes studied. Additionally, a second type of spectrophotometric titration, where a constant concentration of DNA and a gradual increase of [CoCl_2_(dap)_2_]Cl were maintained, was prepared to check the possibility of interaction between this coordination compound and DNA (Appendix A). After each addition of a different amount of DNA solution, electronic absorption spectra were recorded (Appendix A). The calculations were always made from the experimental titration data measured in the absence of any precipitate in the solution.

#### 2.9.2. DNA Cleavage Study

Agarose gel electrophoresis was used to evaluate the Co (III) complexes’ DNA cleavage ability. We used plasmid pBR322 isolated from *E. coli* DH5α by Plasmid Midi AX (A&A Biotechnology, Gdańsk, Poland). The concentration of freshly prepared DNA was measured with a spectrophotometer (Infinite M200 PRO by Tecan, Männedorf, Switzerland) using an absorbance at a 260 and 280 nm value according to the solvent buffer (Tris-HCl/NaCl 5 mM/50 mM). Dilution in the geometric progression of the tested compounds were prepared in the final concentration ranged from 1000 to 15.6 µg/mL and were mixed with the DNA at a constant concentration (62.5 µg/mL). The mixtures of DNA and Co(III) complexes were incubated at 37 °C for 2 h. To enhance the DNA cleaving ability by the complexes, H_2_O_2_ (100 µM) was added to each sample (two concentrations were examined, 500 and 62.5 µg/mL of complexes) and incubated for 5, 10, 15, 30, 60, and 120 min in 37 °C. Controls with DNA and H_2_O_2_ (100 µM) without compounds were also performed. After incubation, the samples were analysed by 0.8% agarose gel electrophoresis in 0.5% TAE buffer for 3 h at 50 V and then detected by a UV illuminator (ChemiDoc^TM^ Touch Imaging System Bio-Rad, Hercules, California, United States).

### 2.10. Statistical Analysis

All experiments were carried out in triplicates, in three independent experimental sets. The means ± SD were used in the statistical analysis of the data and the graphics.

## 3. Results

### 3.1. The Minimum Inhibitory Concentration (MIC) and Minimum Bactericidal Concentration (MBC)

The synthetized and characterized complexes ([CoCl_2_(dap)_2_]Cl (**1**) and [CoCl_2_(en)_2_]Cl) (**2**) were screened in vitro for their antimicrobial activity against a broad spectrum of Gram-positive and Gram-negative bacteria (the reference and clinical strains) using a microbroth dilution method (Table 1). No significant differences in the sensitivity of Gram-positive and Gram-negative bacteria to the tested compounds were observed. The MICs of compounds (**1**) and (**2**) were within the range from 167 to 9333 µg/mL (MBC from 208 to >9333 µg/mL) and were much higher than they were for commercial antibiotics (ampicillin and ciprofloxacin). The most sensitive proved to be S. pyogenes and S. pneumoniae (MIC/MBC for compound (**1**): 229/521 and 208/208 µg/mL; for (**2**): 208/229 and 229/292 µg/mL, respectively), *S. aureus* MSSA 56/AS, *P. acnes* (MIC/MBC for Compound (**1**): 333/458 and 208/208 µg/mL; for (**2**): 583/583 and 229/229 µg/mL, respectively), *C. sporogenes* (MIC/MBC for (**1**): 208/1667 µg/mL; for (**2**): 115/1833 µg/mL), and *H. pylori* (MIC/MBC for (**1**): 167/1833 µg/mL; for (**2**): 208/3000 µg/mL). The most resistant proved to be the strain of *E. faecium* 3844825 (MIC for (**1**): 6833; for (**2**): 9333 µg/mL; MBC: 7333 and >9333 µg/mL, respectively).

In order to examine whether the availability of oxygen affects the activity of cobalt compounds, several bacteria (*S. aureus* ATCC 6538, *S. aureus* MRSA 12673, *E. coli* ATCC 8739, and *E. coli* 12519) were also tested in anaerobic conditions by culturing in GENbaganaer. There were no differences in the activity of the compounds under both conditions.

**Table 1 pharmaceutics-13-00946-t001:** MIC and MBC values in μg/mL of [CoCl_2_(dap)_2_]Cl (**1**), [CoCl_2_(en)_2_]Cl (**2**), ampicillin, and ciprofloxacin against selected bacteria strains.

Strains	[CoCl_2_(dap)_2_]Cl (1)		[CoCl_2_(en)_2_]Cl (2)		Ampicillin		Ciprofloxacin	
	MIC	MBC	MIC	MBC	MIC	MBC	MIC	MBC
*P. aeruginosa* ATCC 9027	1667 ± 516	2333 ± 817	1500 ± 548	2333 ± 817	>2333	>2333	0.67 ± 0.3	0.83 ± 0.3
*P. aeruginosa 12274*	667 ± 258	1500 ± 548	1333 ± 516	1500 ± 548	>2333	>2333	107 ± 33	341 ± 132
*P. mirabilis*	750 ± 246	833 ± 258	1167 ± 408	1333 ± 204	>2333	>2333	21 ± 8.3	19 ± 6.5
*P. mirabilis* 1268	667 ± 258	917 ± 204	1333 ± 516	1333 ± 516	>2333	>2333	17 ± 6.5	21 ± 8.3
*E. hirae ATCC 1052*	2167 ± 983	2333 ± 817	2333 ± 817	2333 ± 817	117 ± 26	192 ± 70	107 ± 33	149 ± 52
*E. faecium* 38344825	6833 ± 2858	7333 ± 1633	9333 ± 3266	>9333	>2333	>2333	85 ± 33	171 ± 66
*E. faecalis* ATCC 51299	2333 ± 817	4667 ± 1633	2667 ± 1033	7333 ± 1633	171 ± 66	171 ± 52	1.33 ± 0.5	1.5 ± 0.6
*E. faecalis* 3937158	1333 ± 516	1733 ± 653	1333 ± 516	2667 ± 1033	>2333	>2333	53 ± 17	171 ± 66
*E. faecalis* 16274	458 ± 102	833 ± 258	1200 ± 408	1600 ± 516	235 ± 52	1195 ± 418	149 ± 52	234 ± 52
*S. aureus* ATCC 6538	375 ± 137	750 ± 433	667 ± 258	1333 ± 516	27 ± 8.3	27 ± 8.3	0.38 ± 0.1	0.58 ± 0.3
*S. aureus* MSSA 56/AS	333 ± 129	458 ± 102	583 ± 204	583 ± 204	1365 ± 529	>2333	0.67 ± 0.3	0.67 ± 0.3
*S. aureus* MRSA 12673	667 ± 258	750 ± 418	833 ± 258	916 ± 204	>2333	>2333	149 ± 52	149 ± 52
*S. aureus* MRSA (hetero-VISA) 6347	458 ± 102	667 ± 258	667 ± 516	667 ± 204	>2333	>2333	0.42 ± 0.1	0.67 ± 0.3
*S. aureus* MRSA N315 (ref.)	667 ± 258	750 ± 204	583 ± 204	583 ± 204	>2333	>2333	1.33 ± 0.5	1.33 ± 0.5
*S. aureus* MRSA 13251	542 ± 225	583 ± 204	833 ± 258	833 ± 258	>2333	>2333	107 ± 33	213 ± 66
*S. aureus* MRSA 15732	458 ± 102	833 ± 258	917 ± 204	917 ± 204	>2333	>2333	171 ± 66	171 ± 52
*S. aureus MRSA 13318*	667 ± 258	833 ± 258	458 ± 102	833 ± 258	>2333	>2333	149 ± 52	149 ± 33
*K. pneumoniae* ATCC 700603	1333 ± 516	1667 ± 516	1667 ± 516	3000 ± 1095	853 ± 264	1365 ± 529	43 ± 16	107 ± 66
*K. pneumoniae* 16205	750 ± 274	667 ± 258	750 ± 274	833 ± 258	1877 ± 418	>2333	85 ± 33	149 ± 52
*K. pneumoniae* 12828	667 ± 274	1167 ± 408	1333 ± 516	1333 ± 204	1877 ± 418	>2333	43 ± 17	171 ± 66
*E. coli* ATCC 8739	833 ± 258	1083 ± 491	1500 ± 548	1833 ± 408	107 ± 33	384 ± 140	0.23 ± 0.08	0.84 ± 0.3
*E. coli* 12519	667 ± 258	833 ± 258	1167 ± 408	1333 ± 516	>2333	>2333	13 ± 4.1	171 ± 66
*E. coli* 12293	833 ± 258	1667 ± 516	1667 ± 516	2333 ± 817	>2333	>2333	21 ± 8.3	107 ± 33
*S. marcescens 12795*	583 ± 333	1333 ± 516	917 ± 258	917 ± 204	149 ± 52	213 ± 66	0.19 ± 0.07	0.21 ± 0.07
*S. marcescens* 13148/2	667 ± 258	1833 ± 408	917 ± 204	1333 ± 516	341 ± 132	427 ± 132	0.33 ± 0.1	0.33 ± 01
*C. sporogenes*	208 ± 65	1667 ± 516	115 ± 25	1833 ± 408	0.833 ± 0.3	1.17 ± 0.4	0.83 ± 0.3	1.17 ± 0.4
*Propionibacterium acnes*	208 ± 65	208 ± 65	229 ± 51	229 ± 51	171 ± 66	171 ± 66	0.75 ± 0.3	1.13 ± 0.5
*S. epidermidis* ATCC 1499	667 ± 258	1083 ± 492	1667 ± 516	1667 ± 516	21 ± 8.3	21 ± 6.5	0.21 ± 0.06	0.42 ± 0.1
*S. epidermidis* MRSE 13199	917 ± 204	1833 ± 1602	1833 ± 408	1833 ± 408	683 ± 264	>2333	0.19 ± 0.07	0.19 ± 0.07
*Salmonella enterica*	1333 ± 516	1667 ± 516	1833 ± 408	2333 ± 817	341 ± 132	>2333	48 ± 18	170 ± 66.1
*Helicobacter pylori*	167 ± 65	1833 ± 408	208 ± 65	3000 ± 1095	3.33 ± 1.03	5.3 ± 2.1	0.67 ± 0.3	1.17 ± 0.4
*S. pyogenes*	229 ± 51	521 ± 26	208 ± 65	229 ± 51	1.7 ± 0.52	4.7 ± 1.6	6 ± 1.9	24 ± 8.8
*S. pneumoniae*	208 ± 65	208 ± 65	229 ± 51	292 ± 102	1.8 ± 0.41	1.8 ± 0.5	11 ± 4.1	-

### 3.2. Serial Passages Assay

In order to assess the increase in bacterial resistance to tested compounds, a serial passages assay was performed using four strains of bacteria (*S. aureus* ATCC 6538, *S. aureus* MRSA 6347, *P. aeruginosa* ATCC 9027, and *P. aeruginosa* 12274) in a medium supplemented with (**1**) and (**2**) below their active concentrations (Table 2 and Figure 2). Each MIC value of Compounds (**1**) and (**2**) was determined by the microbroth dilution method. There was no significant increase in MIC values after 20 passages, both for Compounds (**1**) and (**2**). After 5 passages, there was no increase in the MIC value in any case (the MIC value was the same as the initial value), and after 10 passages, the MIC value doubled only in the case of *S. aureus* MRSA 6347 for Compound (**2**). The following reduction (two times) of susceptibility was observed for *S. aureus* MRSA 6347 and *P. aeruginosa* 12274 after 15 passages for Compound (**1**); the same results were found in the case of *P. aeruginosa* ATCC 9027 for Compound (**2**). After 20 passages, an approximately two-fold reduction in sensitivity was obtained for *S. aureus* ATCC 6538 (Compounds (**1**) and (**2**)) and for *P. aeruginosa* 12274 in the case of Compound (**2**). The MIC value for *S. aureus* MRSA 6347 increased by about three times in comparison to the initial MIC. The presented increases in bacterial resistance to the test compounds were insignificant.

### 3.3. Synergy Assay

The antimicrobial activity of selected antibiotics (ampicillin, gentamicin, ciprofloxacin, polymyxin B, and nalidixic acid) in combination with compounds (**1**) and (**2**) was also determined against both Gram-positive (*S. aureus* ATCC 6538 and *S. aureus* MRSA 6347) and Gram-negative (*P. coli* ATCC 8739 and *E. coli* 12519) bacterial strains by the checkerboard titration method (CSLI) using MH broth in 96-well microtiter plates. Results are shown in Table 3. The significant reductions in MIC values are noted in the case of the combination of compounds (**1**) and (**2**) with nalidixic acid. The mixture of nalidixic acid and compound (**1**) against *S. aureus* MRSA 12673 showed a reduction of MIC values by 8 times (from 448 to 56 µg/mL), and that against *E. coli* 12519 reduced by 4 times (from 1792 to 448 µg/mL), compared to nalidixic acid alone. An eight-fold reduction in MICs was also observed for the combination of the aforementioned antibiotic and Compound (**2**) against *S. aureus* ATCC 6530 and an almost 6-fold reduction in the case of *E. coli* 12519 (from 56 to 7.0 and from 1792 to 320 µg/mL, respectively). On the other hand, there was no reduction in MIC values when nalidixic acid + (**1**) and nalidixic acid + (**2**) mixtures were used against *E. coli* ATCC 8739.

Fractional inhibitory concentration index (FICi) values were also calculated and were in the range of 0.5–4.0 in most cases. These values indicate that there is no interaction in the mixture between the components of antibiotics and either (**1**) or (**2**) (FICi values of >0.5 and ≤4 reflect indifference), even though fold reductions in MIC values for both antimicrobials were observed in some cases (Table 3). The exception is the combination of polymyxin and (**1**) for *E. coli* 12519, where the FIC is 5.0, which would indicate an antagonistic effect.

### 3.4. Microscopic Analysis

#### 3.4.1. Transmission Electron Microscopy

Tested bacterial strains treated for 18 h with MIC concentrations of compounds (**1**) and (**2**) were subjected to transmission electron microscopy, observed in solution and paraffin-embedded bacteria, to visualize their effect on bacterial cell morphology. After incubation in the presence of compounds (**1**) and (**2**) at the MIC level, the shape and morphology of the bacterial cells were distorted. The regular shape of the control cells was clearly visible (Figure 3A,B, control). The images of the bacterial cells treated with the tested compounds, prepared for observation in solution, showed disordered and structural disorganization within the cytoplasm (Figure 3A, +(**1**) and +(**2**)). Some cells turned from the normal round shape into irregular shapes presenting broken and lysed cells. Moreover, the cells coated probably with tested compounds were also seen (black arrows) (Figure 3A, +(**1**)). In the case of paraffin-embedded control bacteria, a regular shape and a cytoplasm of appropriate density inside the cells were observed (Figure 3B, control), whereas an alternation of the intracellular masses and obvious cytoplasmic clear zones were found in the bacterial cells after treatment with compounds (**1**) and (**2**) (Figure 3B, +(**1**), +(**2**)). Visible changes in bacterial cell morphology can indicate that cell membrane permeability was disrupted.

#### 3.4.2. Scanning Electron Microscopy

To visualize the effect of compounds (**1**) and (**2**) on bacterial cell morphology, scanning electron microscopy was carried out. *S. aureus* ATCC 6538 and *P. aeruginosa* ATCC 9027 cells were treated for 24 h with tested compounds at an MIC concentration. The SEM images of untreated cells of bacteria show a well-defined shape free of visible distortions (Figure 4, control). Treatment with both compounds induced remarkable changes in the morphology of the cells in comparison to the control probe (Figure 4, +(**1**) and +(**2**)). Deformation and altered cell morphology were observed in both cases. Cells of different sizes for the *S. aureus* strain and collapsing cells for the *P. aeruginosa* strain were observed. The surfaces of *P. aeruginosa* cells treated with the MIC level of both compounds were wrinkled and irregular. In the case of the microphotograph of *S. aureus,* malformed cells were observed. Moreover, most of the bacterium were ruptured to different degrees.

#### 3.4.3. Fluorescent Microscopy Assay

To evaluate whether Co(III) complexes are able to penetrate bacterial cells, a fluorescent microscopy assay was performed. For these tests, [CoCl_2_(dap)_2_]Cl was selected as a more reactive compound. *P. aeruginosa* ATCC 9077 and *S. aureus* ATCC 6538 were incubated with FITC-labelled [Co(dap)_2_]Cl_2_ for 1 h at 37 °C. Localization of the fluorescence complex in the bacterial cell was visualized using a confocal microscope from Leica Microsystems. Fluorescein conjugated with compound (**1**) ([Co(dap)_2_FLU]Cl_2_) in subinhibitory concentrations (subMIC) penetrated the bacterial cell membrane and has been found to be evenly distributed inside bacterial cells (Figure 5) in the case of both bacteria. Cells of both bacteria in the presence of fluorescein alone did not show any fluorescence.

### 3.5. Interactions of Compounds *(**1**)* and *(**2**)* with DNA

#### 3.5.1. Binding Studies in the Presence of *E. coli* DNA

The binding properties of the diamine complexes of Co(III) with DNA (pBR322) were studied using electronic absorption spectroscopy. To compare quantitatively the binding strength of the two complexes, the intrinsic binding constants, *K*_b_, of the complexes studied with DNA extracted from bacteria were obtained by monitoring the changes in absorbance at 520 nm for complex (**1**) and 510 nm for complex (**2**). The absorption spectra of the free cobalt(III) complexes, treated also as precursors and of their adducts with DNA (at a constant concentration of the compounds) are given in Figure 6 for complexes (**1**) and (**2**), respectively. The UV–vis spectra of Co(III) complexes (**1**) and (**2**) at different contents of DNA show that, in the visible region, the absorption peaks of these solutions showed moderate shifts towards longer wavelengths, which indicates that complexes (**1**) and (**2**) interact with plasmid pBR322 DNA. However, the detailed analysis and comparison of results received for both titrations suggest that Co(III) complex (**2**) with ethylenediamine ligands interacts much more strongly than the analogical Co(III) complex (**1**) with 1,3-diamminepropane ligands inside.

In order to quantitatively compare the ability and strength binding of both complexes with DNA independently, the binding constant (*K*_b_) was determined by calculation based on the changes in the absorbance of the π→π* (261 nm) peaks for (**1**) and of d-d transitions (616 nm) for complex (**2**) after the subsequent addition of amounts of DNA in Tris-HCl solutions. The linear relationships received for both complexes studied are presented in Figure 7 and are according to the Wolfe–Shimer equation [42] below:

$$\frac{\left[\mathrm{DNA}\right]}{\varepsilon_{a}-\varepsilon_{f}}=\frac{\left[\mathrm{DNA}\right]}{\varepsilon_{b\;}-\varepsilon_{f}}+\frac{1}{K_{b\;}\left(\varepsilon_{b}-\varepsilon_{f}\right)}$$ where:

[DNA] is the concentration of DNA (*E. coli*),

ε is the appropriate extinction coefficients:

ε_a_ = A_obsd_/[complex]

ε_f_ = for the free complexes (**1**) or (**2**)

ε_b_ = for compounds (1) or (2) in the fully bound with DNA form, respectively.

The results obtained for complex (**1**), [CoCl_2_(dap)_2_]Cl, from UV–vis titrations suggest that a weak binding mode of interactions between the studied molecules resulted in low hypochromicity with a red shift. Admittedly, we observed wavelengths at 434 nm, whose light absorption by the test mixed solution remained constant. The aforementioned regions discovered suggest that there could be a balance between the components in the solution, but interactions between (**1**) and DNA are rather due to the electrostatic binging mode.

In our opinion, the results for the interactions of complex (**2**), [CoCl_2_(en)_2_]Cl, with DNA of bacteria suggest that it can be assigned as an intercalation type of process. This mode of action is presented by a decrease in absorbance (hypochromism), registered here at 600 nm together with an isosbestic point (575 nm), indicating the equilibria in the medium studied. The value of the binding constants equal to 8.0·10^5^ M^−1^ for the studied Complex (2)–DNA system suggests that this intercalation mode is quantitatively more competitive with groove binding.

#### 3.5.2. DNA Cleavage Study

Agarose gel electrophoresis was used to evaluate the DNA cleavage ability of compounds (**1**) and (**2**). Different amounts of (**1**) and (**2**) complexes were mixed with a fixed amount of supercoiled pBR322 DNA in the medium of 5 mM Tris-HCl, 50 mM NaCl buffer, pH 7.4, in the presence and absence of H_2_O_2_ and incubated for 5, 10, 15, 30, 60, and 120 min at 37 °C. The cleavage patterns of the Co(III) complexes are shown in Figure 8. Control line (DNA in buffer, c) showed a non-cleaved plasmid. The addition of hydrogen peroxide contributed to the active cleavage of the DNA molecule and the appearance of a linear conformation (Form III) of DNA (Figure 8B,C). Compound (**1**) (Figure 8A) did not contribute to a visible change in DNA conformation, and a slight strengthening of the band corresponding to the linear form of DNA (Form III) was observed (Figure 8A). Complex (**2**) (Figure 8A) actively cleaved the plasmid DNA at higher concentrations (1000 and 500 µg/mL). Additionally, DNA retention in the gel well was also observed (Figure 8A, 1000 µg/mL). Furthermore, both complexes cleaved DNA more efficiently in the presence of an oxidant, H_2_O_2_, as compared to the control DNA. After 5 min, at a higher concentration of the tested compounds (500 µg/mL), the bands corresponding to the supercoiled form (Form I) of DNA were invisible, and after 2 h, only smears were observed (Figure 8B). The addition of compound (**1**) at a lower concentration (62.5 µg/mL) resulted in the cleavage of DNA into nicked (Form II) and linear forms after 5 min (Figure 8C), while compound (**2**) cleaved DNA into Forms II and III after 10 min (Figure 8C). After 2 h, the DNA completely disappeared from the gel for both Co(III) complexes (Figure 8C). This means that the DNA was damaged.

## 4. Discussion

### 4.1. The Minimum Inhibitory Concentration (MIC) and Minimum Bactericidal Concentration (MBC)

In the presented study, two Co(III) complexes with a diamine chelate ligands (en—ethylenediamine; dap—1,3-diaminopropane) were synthesized, and the complete physicochemical as well as biological profiles of both compounds were established. A previous report using a broad spectrum of reference and clinical strains of *Candida* spp. showed strong antifungal activity, especially against *C. glabrata* (MIC 16 µg/mL) [5]. The similar results presented Dias et al. (2020) against *Candida glabrata* strains treated with Co(II) complexes with thiocabamoyl–pyrazoline ligands with an MIC from 3.9 to 15.62 µg/mL [24]. In this research, tested compounds were shown to present anti-bacterial activity against a broad spectrum of bacteria, and no significant differences in the sensitivity of the Gram-positive and Gram-negative bacteria to the test compounds were observed. The bacterial strains *S. pyogenes*, *S. pneumoniae*, *P. acnes*, *C. sporogenes*, and *H. pylori* were assessed as the most sensitive. The most resistant proved to be the strain of *E. faecium* 3844825 (Table 1). Studies performed by Dwyer and Sargeson (1959) concerning an evaluation of the antimicrobial activity of the optical isomers of Co(III) with ethylenediamine as N,N-bidentate ligand [Co(en)_3_](NO_3_)_3_ compared with other metal ion complexes revealed that they exhibit high antimicrobial activity, irrespective of the isomer used [27]. Mishra et al. (2008) also showed strong antibacterial properties of the coordination compounds of cobalt(III) with pyridine-amide tri- or bidentate ligands for *P. aeruginosa*, *E. coli*, *S. flexneri*, and *K. planticola* [28].

In our work, the MIC values for Co(III) complexes against bacteria (*Streptococcus* spp., H. *pylori*, C. *sporogenes*, or *P*. *acnes*) grown under anaerobic and microaerophilic conditions were the lowest. To evaluate whether the respiration conditions affect the MIC values of tested complexes, several relatively anaerobic bacterial species, reference and clinical strains of *S. aureus* and *E. coli*, were examined under anaerobic conditions. The obtained results illustrated the lack of influence of the oxygen accessibility in the culture on the anti-bacterial activity of the Co(III) compounds, as compared to studies under aerobic conditions, as MIC results have not changed. However, more tests need to be carried out on a broader spectrum of bacteria strains. Suller and Lloyd (2001) showed in their work that MIC values of vancomycin against *S. aureus* were similar whether tested aerobically or anaerobically [43]. In turn, Oliveira et al. (2019) showed that the Schiff-base ligands and their silver(I) and bismuth(III) complexes were inactive against Gram-positive and Gram-negative aerobic bacteria but were highly active against anaerobic strains. The authors suggested that the mode of action of the tested compounds probably involves the anaerobic reduction of the nitro group by the microorganisms, with a formation of metabolites that are toxic to the tested microorganisms [44].

### 4.2. Serial Passages Assay

We also accessed multi-step resistance studies to evaluate antimicrobial resistance development. Multi-step resistance studies allow for the determination of the effect of the selective pressure of antimicrobials on bacteria, causing the acquisition of mutations at the genetic level [45,46]. These studies involve exposing the bacteria to the sub-inhibitory concentration of antimicrobial agents over many passages, allowing for the formation of resistance mutation over time [47]. In this study, 20 consecutive passages were performed with four strains of bacteria, *S. aureus* ATCC 6538, *S. aureus* MRSA 6347, *P. aeruginosa* ATCC 9027, and *P. aeruginosa* 12274, cultured in a medium supplemented with Compounds (1) and (2) below their MIC values. No significant reduction in the tested bacterial susceptibility to Co(III) compounds was observed. For these compounds, the active concentration after 20 passages was two (*S. aureus* ATCC 6538) or four (*S. aureus* MRSA 6347) times as high as the initial MIC. Only in the case of *P. aeruginosa* for Compound (1) did the MIC value remain at the same level as the initial MIC. The studies assessing bacterial resistance by Mohammad et al. (2017) showed that MIC values for ciprofloxacin against *S. aureus* MRSA increased sevenfold after 14 passages [48]. The study by D’Lima et al. [49] showed that MIC values for reference strains of *S. aureus* ATCC 29213 and *P. aeruginosa* PAO1 for ciprofloxacin after 25 passages increased by about 32 times and for *E. coli* ATCC 25922 by 256 times.

### 4.3. Synergy Assay

The synergy assay was performed for the combination of compounds (**1**) and (**2**) with selected antibiotics (ampicillin, gentamicin, ciprofloxacin, nalidixic acid, and polymyxin B) and showed significant reductions in MIC values only in the case of the mixture of the tested complexes with nalidixic acid. Co(III) complexes with diamine chelate ligands enhance the activity of the aforementioned antibiotic. Nalidixic acid (1-ethyl-1,4-dihydro-7-methyl-4-oxo-1,8-naphthiridin-3-carboxylic acid) is a first-generation quinolone antibiotic used for the treatment of urinary tract infections. This antibiotic selectively and reversibly blocks DNA replication in bacteria by inhibiting a subunit of DNA gyrase and topoisomerase IV. The enzyme is inhibited by trapping the cleavage complex (drug–enzyme–DNA complexes) in which the DNA is broken [50]. It is not clear why the addition of compounds (**1**) and (**2**) increases the activity of nalidixic acid. It is possible that the tested compounds attach to the cleavage complexes or affect the DNA, thus enhancing the effects of the drug.

### 4.4. Microscopic Analysis

The effect of Co(III) complexes with diamine chelate ligands on the bacterial cell were also studied by confocal, scanning electron, and transmission electron microscopes. Using the scanning electron microscope, we observed deformations in the morphology of both tested bacteria, *S. aureus* and *P. aeruginosa*, in the presence of compounds (**1**) and (**2**), while the untreated cells were seen intact. Similarly, in the transmission electron microscope analysis, the bacterial cells showed a distorted shape and morphology. Moreover, we observed irregular shapes of the cells, broken cells, and structural disorganization within the cytoplasm. The described changes in bacterial cells may indicate that cell membrane permeability was disrupted. This was confirmed by analysis using a confocal microscope with the newly synthetized photosensitive complex of the type [Co(dap)_2_FLU]Cl_2_, where the luminescence of the cells was observed, suggesting cell membrane permeability.

### 4.5. Interactions of Compounds *(**1**)* and *(**2**)* with DNA

In this research, we explored the biomolecule affinity of Co(III) diamine complexes by UV–vis titrations. Small molecules, as in our case, can interact selectively with DNA by intercalation and/or groove binding, but the interaction preferences depend on the structure of the DNA-interacting molecules and the nature of the DNA. Small differences in the structure of Co(III) homological complexes may affect the binding types and stability of the complex/DNA adducts. Co(III) complex (**2**), containing a shorter hydrocarbon chain than in the case of complex (**1**), is more favourable for intercalation, in which the moiety can slide between the adjacent base pairs and facilitate the intercalation. The intercalation mode was interpreted in the literature as enhanced π→π* stacking interactions between the aromatic ring of the chromophores and the DNA-base pair’s red shift of the peak to a longer wavelength (bathochromism) [51], which we observed during our spectral intermolecular detection between the complex [CoCl_2_(en)_2_]Cl (**2**) and the DNA. The value of the binding constants equal to 8.0·10^5^ M^−1^ for the studied complex (**2**)–DNA system suggests that this intercalation mode is quantitatively more competitive with groove binding. It can be certainly related to the ionization form of (2) formed as a result of the aquatation process, a form that exists at pH 7.4 to interact with the biomolecule [35]. The value of K_b_ obtained for interactions of the (**1**) complex with DNA is definitely much lower than those above (0.75 × 10^3^ M^−1^), according to the other interaction mechanism promoted by the Co(III)—dap complex. Liu et al., Zhang et al. and Carter et al. [25,52,53,54] showed that Co(III) complexes with a surfactant can bind to the DNA in different binding modes on the basis of their structure, charge, and type of ligands. For example, complexes of the surfactant contain several methyl groups that bind to the DNA by van der Waals interactions and hydrophobic interactions. Moreover, Kumar and Arunachalam (2008) exhibited that surfactant-Co(III) complexes, containing ligands ethylenediamine, triethylenetetramine, 2,2′-bipyridyl, or 1,10-phenantroline, have electrostatic interactions, van der Waals interactions, and/or partial intercalative binding [32]. According to our results, the bacterial DNA can be considered as a cellular target for both complexes studied, susceptible to the formation of adducts with the small Co(III)-diamine complexes (**1**) and (**2**).

The cleaving ability (metallonuclease activity) of complexes (**1**) and (**2**) with supercoiled pBR322 DNA by gel electrophoresis was also performed. The experiment was carried out in the presence and absence of hydrogen peroxide as an oxidant. Supercoiled DNA cleavage is controlled by the relaxation of supercoiled circular conformation of pBR322 DNA in nicked, circular, and/or linear conformations. In our study, complex (**2**) effectively cleaved the DNA in a higher concentration. The same effect was obtained by Thamilarasan et al. (2016) with a higher concentration of the complexes ([Co(acac)(bpy)(N_3_)_2_·H_2_O, ([Co(acac)(en)(N_3_)_2_ and ([Co(acac)(2-pic)(N_3_)_2_, where acac = acetylacetone, bpy = 2,2′-bipyridine, en = ethylenediamine, 2-pic = picolylamine, and NaN_3_ = sodium azide), and the cleavage was found to be much more efficient [55]. This suggested that the cleavage of pBR322 DNA depends on the concentration of the complexes. The different DNA cleavage efficiency of complexes (**1**) and (**2**), but especially (**2**), may be due to the different binding affinity of the complexes to DNA [56,57], as it has been demonstrated by UV–vis titrations experiments. Moreover, the addition of hydrogen peroxide to the DNA + complex (**1**)/(**2**) mixture contributed to the active cleavage of the DNA molecule after 5 min and caused its degradation after 2 h. This may be attributed to the formation of hydroxyl free radicals. Mahalakshmi and Raman (2013) also proposed the occurrence of this phenomenon in their work [58]. They presented that N-(4-aminophenyl)acetamide-derived Schiff-base mixed ligand complexes with Mn(II), Co(II), Ni(II), Cu(II), and Zn(II) show nuclease activity in the presence of oxidant H_2_O_2_, which may be due to the increased production of hydroxyl radicals. The scientific literature indicates that the cleavage of double-stranded DNA by small metal complexes involves two mechanisms: oxidative cleavage and hydrolytic cleavage. The first one to initiate the cleavage requires light and oxidative (leads to the formation of reactive oxygen species, such as ^1^O_2_, O_2_¯, and OH˙) and/or reductive agents [59,60,61,62,63,64,65,66,67], whereas the hydrolytic mechanism requires a *cis*-nucleophile activation to the cleavage of phosphodiester bonds in DNA [68,69,70,71,72,73]. The results of our studies suggested an oxidative mechanism of cleavage with the formation of hydroxyl free radicals. 

The studies aiming to explain the metallo-pharmaceutical action mechanisms are still intensively developing [74,75,76]. Interestingly, a significant number of reports about metallo-drugs concern the structure–activity relationship (SAR) determination together with the molecular target(s) assignation. The complexes studied in our work contained short *N,N*-donor organic ligands and simple monodentate inorganic ligands (en—ethylenediamine; dap—1,3-diaminopropane) in their composition. The exact mode of action of these compounds is still unexplained. In our previous work on *Candida* species, we suggested that tested complexes can change membrane permeability and damage the mitochondrial membrane or the membrane of the endoplasmic reticulum [5]. In the presented study we observed, using TEM and SEM, visible changes in bacterial cell morphology, and the confocal microscope showed the luminescence of cells as well. These results may indicate a disruption of cell membrane permeability. Tümer et al. (1999) suggested that complexes containing ligands with the N and O donor system might be responsible for the inhibition of enzyme production, since enzymes requiring a free hydroxy group for their activity seem to be particularly sensitive to deactivation by the ions of the complexes [77]. The partial sharing of the central ion positive charge with the donor groups and the possible p-electron delocalization within the whole chelate ring decreases the polarity of the molecule, which in turn causes an increase in the lipophilicity of the complex, which enhances the penetration of the complexes into lipid membranes [19] and the blocking of the metal binding sites in the enzymes of the microorganism. Notably, both Co(III) complexes studied in this work are hydrophilic, low-molecular mass compounds. Indeed, they are polar with high water solubility, and their molar mass does not exceed 300 g·mol^−1^. In our opinion, these properties contribute to the lower antibacterial activity. We also hypothesize that the lower antibacterial activity observed for both diamine-chelate Co(III) compounds are due to the active transport out (efflux pumps) of the bacteria cells. Researchers suggest that efflux pumps may be used by the cell as a first-line defense mechanism, avoiding the drug to reach lethal concentrations [78,79].

In summary, special attention was paid to the characterization of coordination compound action modes and to the interactions’ strength. According to the issues discussed above, the possibilities of interactions between complexes (**1**) or (**2**) and the bacterial DNA were planned and examined, and are discussed in detail herein. Studies assessing the interaction of the Co(III) complexes with pBR322 showed a strong mode of interaction in the case of complex (**2**), suggesting an intercalation mode of process, whereas in the case of complex (**1**), these interactions were much weaker and instead indicated an electrostatic binging mode. Kumar and Arunachalam (2008) also proposed the binding of Co(III) complexes with surfactants to DNA as a mode of action of these kind of compounds [29]. Thus, the mode of action of Co(III) complexes with diamine chelate ligands against bacterial cells can be complex, and more studies are necessary to understand the effect of these compounds on bacteria.

## 5. Conclusions

In this study, we evaluated the antibacterial activity of Co(III) complexes with a simple bidentate inorganic ligands (en = ethylenediamine; dap = 1,3-diaminopropane) on a broad spectrum of Gram-positive and Gram-negative bacteria. The analyzed compounds revealed lower activity against bacteria than tested earlier fungi, and no significant differences in the sensitivity of the Gram-positive and Gram-negative bacteria to the test compounds were observed. We hypothesized that the lower activity of compounds against bacteria from the difficulties of overcoming the barrier created by the lipid membranes of bacterial cells, which hinders possible reactions with the target(s) inside the bacterial cell. We also hypothesize that the lower antibacterial activity observed for both diamine-chelate Co(III) compounds, especially for *Pseudomonas aeruginosa* or *Enterococcus* spp., are due to the active transport out (efflux pumps) of the bacteria cells. It was shown that the bacteria grown under anaerobic and microaerophilic conditions (*Streptococcus* spp., H. *pylori*, C. *sporogenes*, or *P*. *acnes*) turned out to be the most sensitive. We proved, by conducting experiments using relatively anaerobic bacteria in anaerobic conditions, that oxygen is not a factor affecting the antibacterial activity of the tested complexes. To evaluate the antimicrobial resistance development, the multi-step resistance studies were performed, and they showed no significant reduction in tested bacterial susceptibility to Co(III) compounds, whereas the synergy assay results revealed significant reductions in MIC values in the case of the mixture of the tested complexes with nalidixic acid. We hypothesized that the aforementioned compounds attach to the cleavage complexes or affect the DNA, thus enhancing the effects of the antibiotic. Indeed, the experiments included the DNA interaction of the Co(III) complexes with plasmid DNA (pBR322) using spectrophotometric titration and DNA cleavage ability of compounds by agarose gel electrophoresis methods revealed that complex (**2**) showed a strong interaction with DNA and indicated an intercalative mode of binding to pBR322. Results for complex (**1**) indicated a much weaker binding mode of interaction between molecules and suggested an electrostatic binding mode. Furthermore, the DNA cleavage ability of the compounds showed, by agarose gel electrophoresis methods, nuclease activity, especially in the presence of oxidant H_2_O_2_ for both complexes. We concluded that this phenomenon may be attributed to the formation of hydroxyl free radicals, but we did not rule out the hydrolytic mechanism of cleavage, highlighting that more tests are needed. Studies using microscopic analysis suggested cell membrane permeability, as evidenced by deformations in the morphology of both tested bacteria.

Finally, we concluded that the mode of action of Co(III) complexes with diamine chelate ligands against bacterial cells can be complex, and a wider spectrum of tests are necessary to understand their antimicrobial effect against bacteria.

## Figures and Tables

**Figure 1 pharmaceutics-13-00946-f001:**
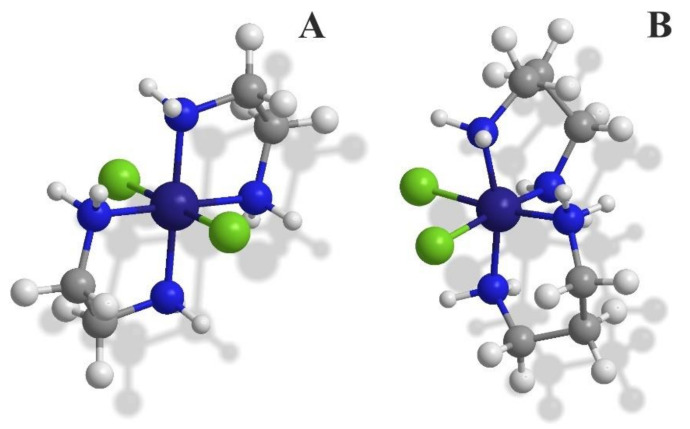
The structures of Co(III) coordination cations adopted from Ref. [5], Front. Microbiol. 2018 presented (**A**) trans-configuration of [CoCl_2_(en)_2_]^+^ (for solid state) and (**B**) cis-configuration of [CoCl_2_(dap)_2_]^+^ (for solution).

**Figure 2 pharmaceutics-13-00946-f002:**
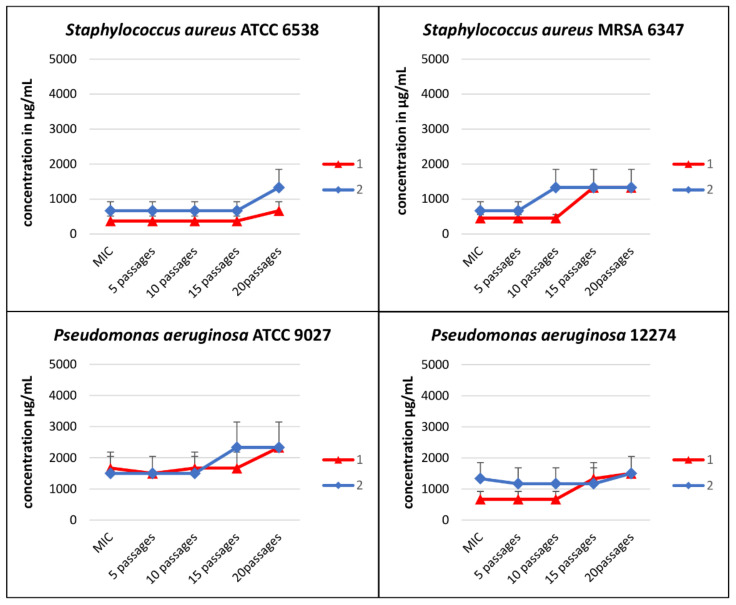
Serial passages assay results for the [CoCl_2_(dap)_2_]Cl (**1**) and [CoCl_2_(en)_2_]Cl (**2**) after 20 subsequent passages of reference and clinical strains of *S. aureus* and *P. aeruginosa.* The compounds below MIC (0.5 × MIC) were used. Data shown are mean ± SD.

**Figure 3 pharmaceutics-13-00946-f003:**
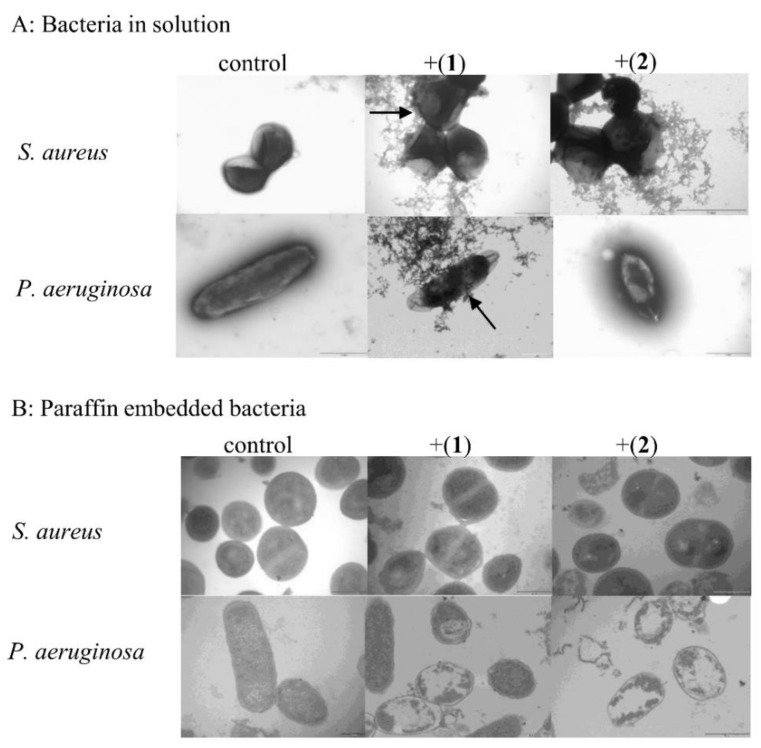
Changes in the cell morphology of *S. aureus* and *P. aeruginosa* cells treated with [CoCl_2_(dap)_2_]Cl- (**1**) and [CoCl_2_(en)_2_]Cl- (**2**). Microscopic preparations prepared in solution and as paraffin-embedded bacteria. Transmission electron microscopy images of *S. aureus* ATCC 6538 and *P. aeruginosa* ATCC 9077 in the absence (control) and presence of Compounds (**1**) (+(**1**)) and (**2**) (+(**2**)). Magnification 20,000×.

**Figure 4 pharmaceutics-13-00946-f004:**
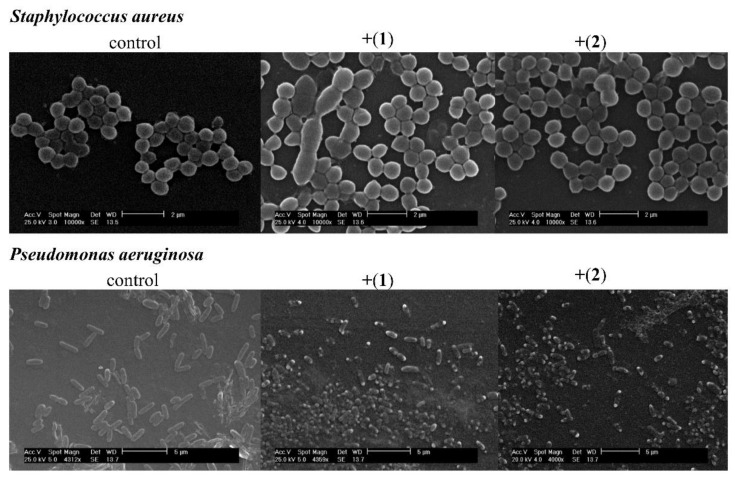
Changes in the cell morphology of *Staphylococcus aureus* and *Pseudomonas aeruginosa* cells treated with [CoCl2(dap)2]Cl- (**1**) and [CoCl2(en)2]Cl- (**2**). Scanning electron microscopy images of *S. aureus* ATCC 6538 (magnification 10000x) and *Pseudomonas aeruginosa* ATCC 9027 (magnification 4000×) in the absence (control) and presence of compound (**1**) (+(**1**)) and compound (**2**) (+(**2**)).

**Figure 5 pharmaceutics-13-00946-f005:**
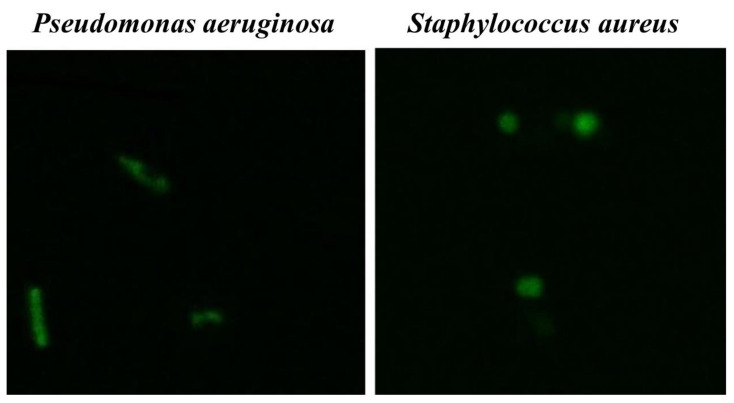
*Pseudomonas aeruginosa* ATCC 9077 and *Staphylococcus aureus* ATCC 6538 with FITC-labeled [CoCl_2_(dap)_2_]Cl. Magnification 2000×.

**Figure 6 pharmaceutics-13-00946-f006:**
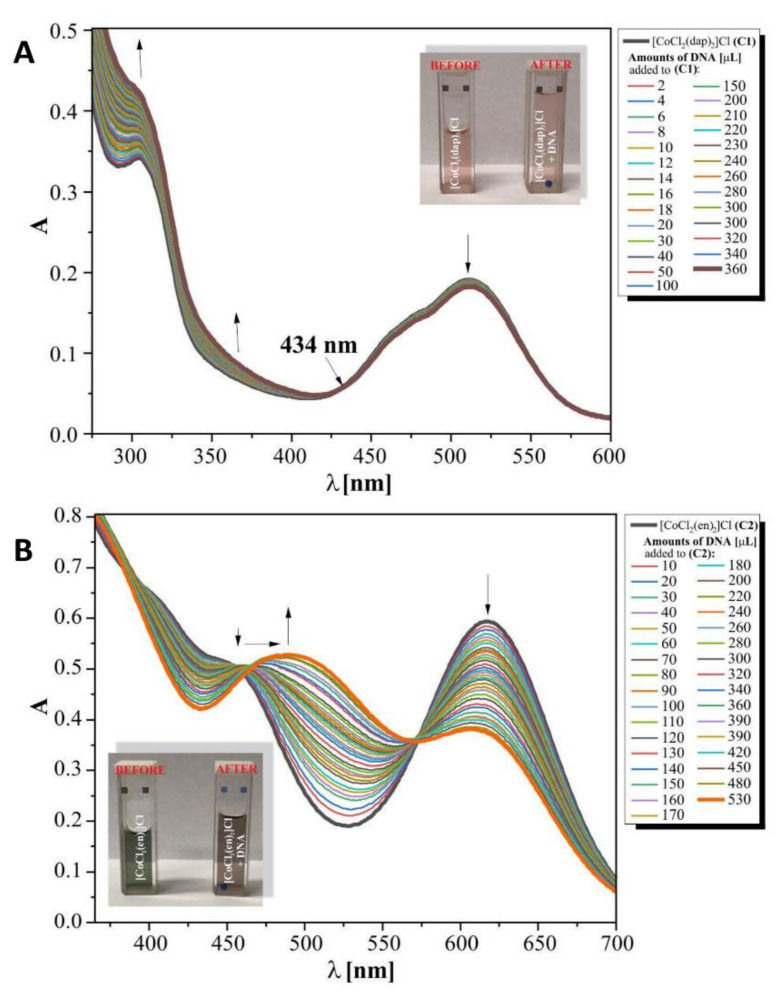
Electronic spectral titration of complex (**1**) and (**2**) with DNA (pBR322) in Tris–HCl buffer, pH 7.39; (**A**): (C1) = 25 mM of compound (**1**), [DNA] = 0.27 mM; (**B**): (C2) = 12 mM of compound (**2**); [DNA] = 0.12 mM. The arrows denote the absorbance changes during the gradual increase of DNA concentration. The photos inside present the colour change of complex (**1**) and (**2**) solutions studied.

**Figure 7 pharmaceutics-13-00946-f007:**
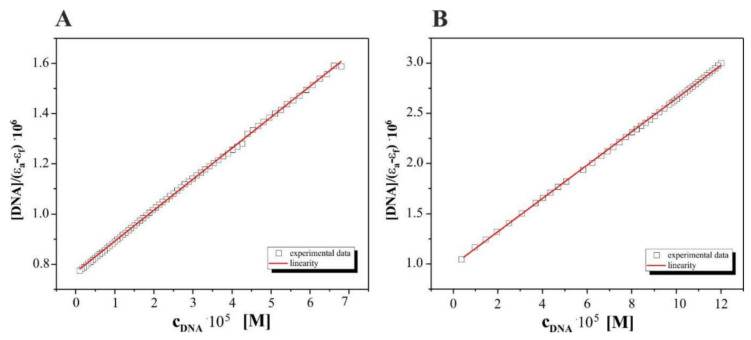
Individual plots of [DNA]/(ε_a_ − ε_f_) vs. [DNA] for the absorption titration of DNA (*E.coli*) with complexes: (**A**). [CoCl_2_(dap)_2_]Cl (**1**) and (**B**). [CoCl_2_(en)_2_]Cl (**2**) studied in Tris–HCl buffer; association constant *K*_b_: 0.75·10^3^ M^−1^ (R = 0.99950, n = 70 experimental points) for (**1**); .*K*_b_: 8.0·10^5^ M^−1^ (R = 0.99987, n = 46 experimental points) for (**2**).

**Figure 8 pharmaceutics-13-00946-f008:**
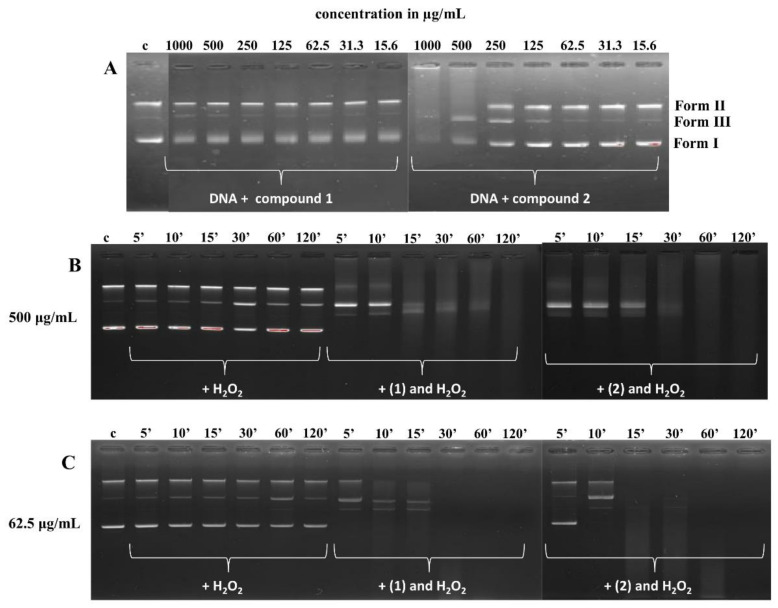
The cleavage of supercoiled pBR322 DNA (62.5 µg/mL) by the complexes at 37 °C in 5 mM Tris HCl, 50 mM NaCl buffer. (**A**)—DNA incubated with compounds (**1**) and (**2**), respectively; c, control line—DNA in buffer (5 mM Tris HCl, 50 mM NaCl) (1:1), 1000–15.6 µg/mL, decreasing concentration of compounds (**1**) and (**2**). (**B**)—DNA incubated with H_2_O_2_ (100 µM), compound **(1)** (500 µg/mL) + H_2_O_2_ (100 µM), and (**2**) (500 µg/mL) + H_2_O_2_ (100 µM), respectively; c, control line—DNA in buffer (5 mM Tris HCl, 50 mM NaCl) (1:1), 5–120 min, time of incubation. (**C**)—DNA incubated with H_2_O_2_ (100 µM), compound (**1**) (62.5 µg/mL) + H_2_O_2_ (100 µM) and (**2**) (62.5 µg/mL) + H_2_O_2_ (100 µM), respectively; c, control line—DNA in buffer (5 mM Tris HCl, 50 mM NaCl) (1:1), 5–120 min, time of incubation.

**Table 2 pharmaceutics-13-00946-t002:** MIC (µg/mL) values in the presence of [CoCl_2_(dap)_2_]Cl (**1**) and [CoCl_2_(en)_2_]Cl (**2**) passaging results for selected bacterial strains.

Strains	Compound	Initial MIC (µg/mL)	MIC after 20 Passages (µg/mL)
*S. aureus* ATCC 6538	[CoCl_2_(dap)_2_]Cl (**1**)	375 ± 137	667 ± 258
	[CoCl_2_(en)_2_]Cl (**2**)	667 ± 258	1333 ± 516
*S. aureus* MRSA 6347	[CoCl_2_(dap)_2_]Cl (**1**)	458 ± 102	1333 ± 516
	[CoCl_2_(en)_2_]Cl (**2**)	667 ± 516	2333 ± 817
*P. aeruginosa* ATCC 9027	[CoCl_2_(dap)_2_]Cl (**1**)	1667 ± 516	2333 ± 817
	[CoCl_2_(en)_2_]Cl (**2**)	1500 ± 548	2333 ± 817
*P. aeruginosa* 12274	[CoCl_2_(dap)_2_]Cl (**1**)	667 ± 258	1500 ± 548
	[CoCl_2_(en)_2_]Cl (**2**)	1333 ± 516	1500 ± 548

**Table 3 pharmaceutics-13-00946-t003:** (**a**) Fractional inhibitory concentration index (FICi) values of [CoCl_2_(dap)_2_]Cl (**1**), ampicillin, gentamicin, nalidixic acid, and polymyxin B. (b) Fractional inhibitory concentration index (FICi) values of [CoCl_2_(en)_2_]Cl (**2**), ampicillin, gentamicin, nalidixic acid, and polymyxin B.

**(a)**									
**The Bacterial Strain**	**Antibiotic**	**MIC of Antibiotics (µg/mL)**		**MIC of (1) (µg/mL)**		**Fold Reduction of** **MIC of Antibiotic**	**Fold Reduction** **of MIC of (1)**	**FIC**	**Interpretation**
		**Alone**	**Com.**	**Alone**	**Com**				
*S. aureus* *ATCC 6530*	Ampicillin	27 ± 8.3	27 ± 8.3	458 ± 102	417 ± 144	-	0.9	2.1	indifference
	Gentamicin	0.438 ± 0.13	0.438 ± 0.13	375 ± 137	375 ± 137	-	-	2	indifference
	Ciprofloxacin	0.38 ± 0.1	0.38 ± 0.1	375 ± 137	375 ± 137	-	-	2	indifference
	Nalidixic acid	53.3 ± 18.5	53.3 ± 18.5	375 ± 137	375 ± 137	-	-	2	indifference
	Polymyxin B	85.3 ± 37	85.3 ± 37	417 ± 144	208 ± 72	-	2	1.5	indifference
*S. aureus* MRSA 12673	Ampicillin	>4096	>4096	667 ± 258	667 ± 258	-	-	-	-
	Gentamicin	0.23 ± 0.08	0.12 ± 0.03	667 ± 258	667 ± 258	1.9	-	1.5	indifference
	Ciprofloxacin	149 ± 52	85.3 ± 37	416.7 ± 144	208 ± 72	1.8	2	1	indifference
	Nalidixic acid	448 ± 128	56 ± 16	667 ± 258	667 ± 258	8	-	1.1	indifference
	Polymyxin B	224 ± 64	224 ± 64	667 ± 258	667 ± 258	-	-	2	indifference
*E. coli* ATCC 8739	Ampicillin	107 ± 37	53.5 ± 28.5	833 ± 258	417 ± 144	2	2	0.7	indifference
	Gentamicin	0.42 ± 0.14	0.22 ± 0.06	437 ± 125	219 ± 63	2	2	1	indifference
	Ciprofloxacin	0.23 ± 0.08	0.12 ± 0.03	900 ± 223	313 ± 125	1.9	4	0.7	indifference
	Nalidixic acid	7.0 ± 2.0	3.6 ± 0.9	900 ± 223	437 ± 125	1.9	2	1	indifference
	Polymyxin B	0.438 ± 0.13	0.89 ± 0.25	875 ± 250	875 ± 250	-	-	3	indifference
*E. coli* 12519	Ampicillin	3584 ± 1024	3584 ± 1024	667 ± 258	667 ± 258	-	-	2	indifference
	Gentamicin	112 ± 32	56 ± 16	667 ± 258	667 ± 258	2	-	1.5	indifference
	Ciprofloxacin	13 ± 4.1	13 ± 4.1	667 ± 258	667 ± 258	-	-	2	indifference
	Nalidixic acid	1792 ± 512	448 ± 128	667 ± 258	219 ± 63	4	3	0.6	indifference
	Polymyxin B	0.112 ± 0.03	0.438 ± 0.13	667 ± 258	667 ± 258	-	-	5	antagonistic
**(b)**									
**The Bacterial Strain**	**Antibiotic**	**MIC of Antibiotics (µg/mL)**		**MIC of (2) (µg/mL)**		**Fold Reduction of MIC of** **Antibiotic**	**Fold Reduction of** **MIC of (2)**	**FIC**	**Interpretation**
		**Alone**	**Com.**	**Alone**	**Com.**				
*S. aureus* *ATCC 6530*	Ampicillin	27 ± 8.3	27 ± 8.3	667 ± 258	667 ± 258	-	-	2	indifference
	Gentamicin	0.438 ± 0.13	0.23 ± 0.08	667 ± 258	667 ± 258	2	-	1.5	indifference
	Ciprofloxacin	0.38 ± 0.1	0.17 ± 0.06	667 ± 258	667 ± 258	2.2	-	1.5	indifference
	Nalidixic acid	56 ± 16	7.0 ± 2.0	667 ± 258	667 ± 258	8	-	1.1	indifference
	Polymyxin B	80 ± 32	80 ± 32	667 ± 258	333 ± 63	-	2	1	indifference
*S. aureus* MRSA 12673	Ampicillin	>4096	>4096	883 ± 258	883 ± 258	-	-	1.2	indifference
	Gentamicin	0.23 ± 0.08	0.12 ± 0.03	438 ± 125	438 ± 125	2	-	1.5	indifference
	Ciprofloxacin	149 ± 52	74.5 ± 29	438 ± 125	225 ± 56	2	1.9	1	indifference
	Nalidixic acid	461 ± 114	115 ± 33	438 ± 125	225 ± 56	4	1.9	0.7	indifference
	Polymyxin B	224 ± 64	224 ± 64	438 ± 125	438 ± 125	-	-	2	indifference
*E. coli* ATCC 8739	Ampicillin	107 ± 33	53.5 ± 19	1500 ± 548	750 ± 274	2	2	1	indifference
	Gentamicin	0.438 ± 0.13	0.12 ± 0.03	1750 ± 500	875 ± 250	4	2	0.7	indifference
	Ciprofloxacin	0.23 ± 0.08	0.23 ± 0.08	1750 ± 500	875 ± 250	-	2	1.5	indifference
	Nalidixic acid	7.2 ± 1.8	7.2 ± 1.8	1750 ± 500	875 ± 250	-	2	1.5	indifference
	Polymyxin B	0.438 ± 0.13	0.22 ± 0.06	1750 ± 500	875 ± 250	2	2	1	indifference
*E. coli* 12519	Ampicillin	3584 ± 1024	3584 ± 1024	1167 ± 408	1167 ± 408	-	-	2	indifference
	Gentamicin	115 ± 29	115 ± 29	1200 ± 447	1200 ± 447	-	-	2	indifference
	Ciprofloxacin	13 ± 4.1	13 ± 4.1	1167 ± 408	600 ± 224	-	1.9	1.5	indifference
	Nalidixic acid	1792 ± 512	320 ± 128	1200 ± 447	1200 ± 447	5.6	-	1.1	indifference
	Polymyxin B	0.109 ± 0.03	0.22 ± 0.06	1167 ± 408	1167 ± 408	-	-	3	indifference

## Data Availability

Not applicable.

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
