# Peer review of "Antibacterial Activity of Co(III) Complexes with Diamine Chelate Ligands against a Broad Spectrum of Bacteria with a DNA Interaction Mechanism"

_pharmaceutics, 2021, doi:10.3390/pharmaceutics13070946_

Round 1
Reviewer 1 Report
Turecka et all present in this work the synthesis of Co(III) complexes and its application as antibacterial agents in a variety of bacterial strains together with its DNA interaction mechanism. The manuscript presents science but some general aspects related to the chemistry presented in this manuscript must be elucidated in deep before publication. Overall, this manuscript is worth of publication in Pharmaceutics due to new findings in biological studies.
- The authors state that the ligands used are monodentate ligands Line 77. Are this ligands really monodentate in the complexes structure?
- The manuscript introduction is well motivated with classical and old references cited. In my opinion, some recent published works involving antifungal, antibacterial and antitumoral studies of new coordination compounds should be cited as Dias et al, J. Inorg. Biochem. 2020 (213) 111277; Dantas et al in Molecules 2018 (23) 1856 and Favarin et al in Inorg. Chim. Acta 2019 (492) 235 and New. J. Chemistry 2020 (44) 6862. Here luminescent coordination compounds also were studied in this context. Some results should be compared with findings is this new manuscript.
- A reactional scheme displaying the reactions of complexes 1, 2 and 3 and its respective ligands must be added in the manuscript. The authors did not show any structural correlation in figures. I think it should be clarified and showed figures also representing the structures of Co(III) complexes obtained.
- Line 92 1,3-diaminepropane not diamminopropone
- In the synthesis of [Co(dap)2FLU]Cl complex some aspects needs to be clarified. Why do the authors used 3:4 stoichiometric ration between cobalt complex and fluorescein sodic salt?
- Why do the authors used NaBr in this reaction?
- Is the complex [Co(dap)2FLU]Cl new? Was this complex published in the literature so far?
- I think this complex needs to be chemically characterized in deep. For example, Co(III) is a diamagnetic specie, thus NMR studies can be performed? HRMS studies can be performed.
- The Complex [Co(dap)2FLU]Cl was also used in Fluorescence microscopy, in this sense, why do the authors have not stuied photopysical properties of this complex in solution? What was the excitation wavelength used to excitation of this complex in bioimaging assays?
- The authors stated (line 320) that “Cells of both bacteria in the presence of fluorescein alone did not show any fluorescence” thus, which kind of synergic effect between metal and ligand is occurring? Again, may be photophysical behavior studies of this complex in solution could help clarify this phenomenon.
Author Response
General Authors’ comment
We thank the Reviewers for their comments and for taking the time to read and review our work. All changes have been included in the revised manuscript version.
We have acquainted of the authorship policies for the Pharmaceutics conform.
Response to Reviewer 1 Comments
We enclose a reference to the reviewers' comments and answers to the questions posed to us, along with a list of changes that have been made in the revised form of manuscript.
Point 1: The authors state that the ligands used are monodentate ligands Line 77. Are this ligands really monodentate in the complexes structure?
Response 1: According to Reviewer #1 comments and related doubts included in the review, the additional measurements were performed to established the proper coordination mode of FLU ligand in the indicator complex denotes in the manuscript by not proper formula [Co(dap)2FLU]Cl. Indeed, the ligand shows bidentate ability to bind the Co(III) ion. Moreover, the lower subscript related to chloride counter ions disappeared in the formula as a result of paper editing. The new version of manuscript includes the proper description of the indicator structure proved by XRD data and parameters, supported by 1H NMR spectra together with the adequate discussion and analysis compared to its precursors structures.
Point 2: The manuscript introduction is well motivated with classical and old references cited. In my opinion, some recent published works involving antifungal, antibacterial and antitumoral studies of new coordination compounds should be cited as Dias et al, J. Inorg. Biochem. 2020 (213) 111277; Dantas et al in Molecules 2018 (23) 1856 and Favarin et al in Inorg. Chim. Acta 2019 (492) 235 and New. J. Chemistry 2020 (44) 6862. Here luminescent coordination compounds also were studied in this context. Some results should be compared with findings is this new manuscript.
Response 2: The mentioned references were included in the new version of the manuscript according to Reviewer #1 suggestions. Additionally, we put some comments directly related to the findings in the cited articles. Interestingly, the luminescent properties of the coordination compounds are considered by us and will be the main research object in the next manuscript as a continuation of those results currently reported in the article.
Point 3: A reactional scheme displaying the reactions of complexes 1, 2 and 3 and its respective ligands must be added in the manuscript. The authors did not show any structural correlation in figures. I think it should be clarified and showed figures also representing the structures of Co(III) complexes obtained.
Response 3: The scientific objects, Co(III) complexes structures are reported in our earlier articles, which were cited in the manuscript text as Ref. No. [35] – pKa investigations and [5] – structures presentations. The lack of the Figures with Co(III) structures published recently was associated with our concerns about the Figures repetition. To avoid such suggestions, we decided to focused the Reader attention on the biological aspects of those research achievements, like Co(III) complexes abilities to interact with DNA biomolecule and the strength of such action.
Point 4: Line 92 1,3-diaminepropane not diamminopropone
Response 4: The correction was made in the revised manuscript.
Point 5: In the synthesis of [Co(dap)2FLU]Cl complex some aspects needs to be clarified. Why do the authors used 3:4 stoichiometric ration between cobalt complex and fluorescein sodic salt?
Response 5: There were many probes to establish the synthesis pathway and reaction conditions to obtain the stable product with the fluorescein located inside the coordination sphere and bounded to Co(III) ion. However, only the use of the mentioned stoichiometric ratio of substrates gave the expected stable coordination product.
Point 6: Why do the authors used NaBr in this reaction?
Response 6: The procedure synthesis was developed long time based on the literature recipes. One of them, suggested to use the addition of the strong electrolyte, what was probably related to keep the constant ionic strength in the solution. Due to above, we chosen simple salt (NaBr) to avoid the additional competitive reactions of substitution during the FLU complexation.
Point 7: Is the complex [Co(dap)2FLU]Cl new? Was this complex published in the literature so far?
Response 7: The procedure synthesis was not reported in the literature until now. The complex is new and its’ structure was proved by additional structural data included in the presented and revised manuscript of the article. The newly obtained results were added to a supplementary file to give the Readers opportunity to focused their attention on the DNA interactions differences in mode of action presented by two structural similar Co(III) coordination compounds.
Point 8:. I think this complex needs to be chemically characterized in deep. For example, Co(III) is a diamagnetic specie, thus NMR studies can be performed? HRMS studies can be performed.
Response 8: Indeed, the newly results about the mentioned complex were included in the revised manuscript to support the conclusions and to present the complete characterization of the Co(III) complex with FLU. NMR studies can be performed according to the thematic literature (Ref. article below*) and was also included in the revised manuscript.
*Mark R. McClure, K.W. Jung, Jay H. Worrell, High-resolution NMR analysis of cobalt(III) complexes with 1,8-diamino-3,6-dithiaoctane, Coordination Chemistry Reviews, 174 (1998) 33–50
Point 9:. The Complex [Co(dap)2FLU]Cl was also used in Fluorescence microscopy, in this sense, why do the authors have not studied photophysical properties of this complex in solution? What was the excitation wavelength used to excitation of this complex in bioimaging assays?
Response 9: The excitation wavelength used to excitation of mentioned complex in bioimaging assay was 490 nm.
Point 10: The authors stated (line 320) that “Cells of both bacteria in the presence of fluorescein alone did not show any fluorescence” thus, which kind of synergic effect between metal and ligand is occurring? Again, may be photophysical behavior studies of this complex in solution could help clarify this phenomenon.
Response 9 and 10: The photophysical properties of the Co(III) coordination compound with FLU as well as some synergy effects will be considered by us in the next manuscript as the main scientific problem to resolve. However, the metal ionic center plays probably the crucial role; is responsible for the stiffening of the entire skeleton of the complex; stimulates photophysical properties of FLU. The synergy mentioned was proposed according to the formation possibility of conjugation bond between M-L (Co/FLU). The coordination chemistry offers new structural solutions together with the specific properties often which are not observed in the case of metal complexes precursors. We will try to reconsider the phenomenon presented in the article and in our opinion the explanation is related with the enantiomers of Co(III)-FLU complex exist.
Cells of bacteria without tested compounds are the control probes containing living cells. Fluorescein is not able to penetrate intact cell membrane. The luminescence of the cells suggests cell membrane permeability.
List of changes in the revised manuscript:
1) Linguistic changes have been made.
2) The XRD results were added and discussed.
3) The NMR analysis were made and also included in the revised manuscript.
4) The new Figures and their description were added in the supplementary file.
5) The formula of Co(III)-FLU complex structure was checked and corrected inside the text (the subscript lack).

Reviewer 2 Report
The Ms submitted by Dr. K. Turecka describes the biological effect of the two complexes [CoCl2(dap)2]Cl (1) and [CoCl2(en)2]Cl (2) against Gram-positive and Gram-negative bacteria as well as DNA interaction with plasmid pBR322 using different biological techniques, which were performed in adequate and scientific way. The Ms lacks the basic knowledge of inorganic chemistry concerning the nature of the two selected complexes. Although in the introduction, it was stated that “the electrochemical redox potential of complex and the structure of the ligand have a significant impact on the effectiveness and the stability of compounds used”, the authors dis not specify the stereochemistry of the two complexes because this is very important in considering the species under the working pH values (pH ~7.4). Yes, it well established that chelation using strong ligand fields increase the stability of Co(III) complexes. The Ms suffers some serious issues:
- Most likely the two prepared dichlorido complexes display “trans geometry”, which are well known to undergo cis/trans isomerization, none of these was reported.
- In addition to the cis/trans isomerization, the two complexes undergo hydrolysis and it is very important to know which hydrolytic products are formed under the biological pH values. Complex 1 is know to rapidly hydrolyzed to [Co(dap)(H2O)2]3+ and this is a very fast process (may be < 1 min; labile), whereas complex 2 is slowly hydrolyzed to [Co(en)2(H2O)Cl]2+ and [Co(en)2(H2O)2]2+. Therefore, the incubation time is very crucial in determining which species exit.
- In addition, the above three aquated species undergo acid dissociation to produce cis-[Co(dap)2(OH)2]+ (please see Chim. Acta 1988, 146, 3) and a mixture of [Co(en)2(H2O)(OH)]2+/ [Co(en)2(OH)2]+. Therefore, which of these species exist at pH = 7.4?
- In the introduction, I suggest the reviewers to take a look to the paper cleavage of DNA promoted by cobalt(III)-tetraamine complexes (Polyhedron 2009, 28, 1221-1228) and some references in. Also for Co2+/DNA cleavage please see: Dalton Trans. 2014, 43, 10086-10103. http://dx.doi:10.1039/C4DT00615A
- The introduction is poorly written and should be REWRITTEN.
- The English of the Ms must be polished and revised carefully.
- The author should address why DNA cleavage by 1 is producing only form II, whereas complex 2 gives a mixture of II and III. In the latter case this may result from the presence of two reactive species in solution [Co(en)2(H2O)(OH)]2+ and [Co(en)2(OH)2]+.
Following all of the above, I can not suggest the publication of this Ms in the present form. Extensive major revision is required and then resubmitted as a new paper.
Author Response
General Authors’ comment
We thank the Reviewers for their comments and for taking the time to read and review our work. All changes have been included in the revised manuscript version.
We have acquainted of the authorship policies for the Pharmaceutics conform
Response to Reviewer 2 Comments
Point 1: Although in the introduction, it was stated that “the electrochemical redox potential of complex and the structure of the ligand have a significant impact on the effectiveness and the stability of compounds used”, the authors dis not specify the stereochemistry of the two complexes because this is very important in considering the species under the working pH values (pH ~7.4).
Response 1: The mentioned aspects were considered by us and resolved in the previous achievements. Below, we decided to give the detailed explanation to Reviewer #2 comments. Interestingly, we checked the redox potentials of Co(III) complexes studied [Ref. No. 35]:
„The values of the reduction potentials were determined by using cyclic voltamperometry technique for two selected coordination compounds. The results showed that the reduction potential for [Co(dap)2(H2O)2](ClO4)3 is 0.997 V and for [Co(en)2(H2O)2](ClO4)3 is 0.611 V. Co(III) complex containing 1,3-diaminopropane is reduced much easier to Co(II) complex, which is shown the number of peaks obtained on cathode, anode and the occurrence of peak values of their potentials.“
Point 2: Most likely the two prepared dichlorido complexes display “trans geometry”, which are well known to undergo cis/trans isomerization, none of these was reported.
Response 2: We agree. The mentioned aspects were taken into account during the experiments performed. Note, that we included in our measurement conditions the time necessary for Co(III) complexes isomerization occurred spontaneously in aqueous buffer solutions (this is why we did not name the kind of isomer cis-trans in the solution). The trans forms of Co(III) – diamine complexes are stable only in solid states.
Point 3: In addition to the cis/trans isomerization, the two complexes undergo hydrolysis and it is very important to know which hydrolytic products are formed under the biological pH values. Complex 1 is know to rapidly hydrolyzed to [Co(dap)(H2O)2]3+ and this is a very fast process (may be < 1 min; labile), whereas complex 2 is slowly hydrolyzed to [Co(en)2(H2O)Cl]2+ and [Co(en)2(H2O)2]2+. Therefore, the incubation time is very crucial in determining which species exit.
Response 3: Interestingly, the mentioned by Reviewer #2 suggestions were considered and calculated by us in the previous our articles old Ref. No. [5] and [35]. Interestingly, we published the values of pKa for both Co(III) complexes together with the comments about their species formed, see old Ref. No. [35]. We added the additional note to focused Reader attention about these aspects of the research. Below some data published as an explanation and to prove that we included all known results in the currently article reported [35]:
“On the basis of potentiometric and spectrophotometric data from titration curves for the complexes of type trans-[Co(N,N)2(H2O)2](ClO4)3 (where N,N=en) and 11 (where N,N=dap), following acid-base equilibrium model was proposed:
[Co(N,N)2(H2O)2]3+ + OH- = [Co(N,N)2(H2O)(OH)]2+ + H2O (2)
[Co(N,N)2(H2O)(OH)]2+ + OH- = [Co(N,N)2(OH)2]+ + H2O (3)
The model proposed was used to prepare the stoichiometric matrix and to determine the values of pK (Table 1).
Table 1. Acidity constants of the Co(III) complexes, obtained by the spectroscopic and potentiometric titration methods
|
Complex |
Spectrophotometric titration |
Potentiometric titration |
|
||||
|
pK1 |
pK2 |
pK1 |
pK2 |
|
|||
|
[Co(en)2(H2O)2](ClO4)3 |
5.74 (± 0.11) |
8.19 (± 0.35) |
5.98 (± 0.23) |
8.08 (± 0.34) |
|
||
|
[Co(dap)2(H2O)2](ClO4)3 |
2.42 (± 0.14) |
8.02 (± 0.17) |
2.65 (± 0.45) |
8.51 (± 0.21) |
|
||
|
2.57 (±0.03)* |
not observed* |
2.47 (± 0.21)* |
not observed* |
||||
*results obtained from titration with KOH as a titrant
Point 4: In addition, the above three aquated species undergo acid dissociation to produce cis-[Co(dap)2(OH)2]+ (please see Chim. Acta 1988, 146, 3) and a mixture of [Co(en)2(H2O)(OH)]2+/ [Co(en)2(OH)2]+. Therefore, which of these species exist at pH = 7.4?
Response 4: The answer were included in the above. Based on the model of acid dissociation as well as the pKa values cited, the Co(III) complexes species at pH 7.4 were detailed defined.
Point 5: In the introduction, I suggest the reviewers to take a look to the paper cleavage of DNA promoted by cobalt(III)-tetraamine complexes (Polyhedron 2009, 28, 1221-1228) and some references in. Also for Co2+/DNA cleavage please see: Dalton Trans. 2014, 43, 10086-10103. http://dx.doi:10.1039/C4DT00615A
Response 5: The mentioned articles were included in the revised manuscript according to the Reviewer #2 suggestion. We agree, that those literature is important and also worth to take a look.
Point 6: The introduction is poorly written and should be REWRITTEN.
Response 6: The Introduction was rewritten in the revised manuscript according to the Reviewer #2 suggestion.
Point 7: The English of the Ms must be polished and revised carefully.
Response 7: Indeed, the language needed to be improved, the corrections of the text were made in the revised manuscript.
Point 8: The author should address why DNA cleavage by 1 is producing only form II, whereas complex 2 gives a mixture of II and III. In the latter case this may result from the presence of two reactive species in solution [Co(en)2(H2O)(OH)]2+ and [Co(en)2(OH)2]+.
Response 8: The above Reviewer #2 comment was considered based on the data included in our earlier article about the pKa determination of Co(III) compounds studied (visit Ref. No. 35 and 5). However, the second deprotonation reaction is resulting to the production of minimal forms mentioned [Co(N,N)2(OH)2]+ and its’ existing could be omitted based on the differences between pKa values like in case of the forms related to hexaaqua metal complexes formed in the aqueous solution.
List of changes in the revised manuscript:
1) Linguistic changes have been made.
2) The XRD results were added and discussed.
3) The NMR analysis were made and also included in the revised manuscript.
4) The new Figures and their description were added in the supplementary file.
5) The formula of Co(III)-FLU complex structure was checked and corrected inside the text (the subscript lack).

Round 2
Reviewer 1 Report
I consider the manuscript suitable for publication in Pharmaceutics after the revisions made for the authors. Of course, some aspects related to the luminescent solution behavior of the complexes need to be clarified in deep in future works for better understanding the photophysical behavior inside the cell. For this work, the lack of these studies in solution does not compromise the quality and the results presented by the authors. In this sense, I recommend this article for publication as it stands.
Thank you for your contribution to this research area.
Author Response
General Authors’ comment
We thank the Reviewers for their comments and for taking the time to read and review our work. All changes have been included in the revised manuscript version.
We have acquainted of the authorship policies for the Pharmaceutics conform.
Response to Reviewer 1 Comments
We enclose a reference to the reviewers' comments and answers to the questions posed to us, along with a list of changes that have been made in the revised form of manuscript.
We are very grateful the Reviewer 1 recommended our article for publication in Pharmaceutics. As Reviewer suggested, in our future work we will focus on the photophysical behavior of luminescent solution inside the cell.
